# Physical Health-Related Quality of Life in Relation to Mediterranean Diet Adherence in a Sample of Greek Asthma Patients: A Pilot Study

**DOI:** 10.3390/jpm13101512

**Published:** 2023-10-20

**Authors:** Lamprini Kontopoulou, Ourania S. Kotsiou, Konstantinos Tourlakopoulos, Georgios Karpetas, Eva V. Paraskevadaki, Foteini Malli, Ioannis Pantazopoulos, Zoe Daniil, Konstantinos I. Gourgoulianis

**Affiliations:** 1Respiratory Disorders Lab, Faculty of Nursing, University of Thessaly, 41500 Larissa, Greece; mallifoteini@yahoo.gr; 2Department of Human Pathophysiology, Faculty of Nursing, University of Thessaly, 41500 Gaiopolis, Greece; 3Department of Respiratory Medicine, Faculty of Medicine, University of Thessaly, 41110 Biopolis, Greece; kntourlakopoulos@gmail.com (K.T.); gkarpetas@uth.gr (G.K.); pantazopoulosioannis@yahoo.com (I.P.); zdaniil@uth.gr (Z.D.); kgourg@med.uth.gr (K.I.G.); 4Research and Statistics Consulting Services, 10681 Athens, Greece; eva.parask@stepupadvisor.gr

**Keywords:** nutrition, asthma, Mediterranean Diet, asthma severity, quality of life, obesity

## Abstract

The role of nutrition in the management of asthma in obese patients is of increasing interest due to their limited response to inhaled corticosteroids. Some studies note that through diet and lifestyle, there can be an improvement in asthma control. The aim of the present study was to investigate the adherence to the Mediterranean Diet and its association with asthma severity and quality of life in patients with bronchial asthma. This is a cross-sectional study of 85 patients (70.6% female), with a mean age of 57 years, from the General University Hospital of Larissa and, more specifically, patients of the outpatient asthma clinic. Data were collected with the use of specific questionnaires. In relation to BMI, 12.9% of participants were of a normal weight, 45.9% were overweight, 25.9% were obese level I, 5.9% were obese level II, and 9.4% were in the morbidly obese range. Based on the Med Diet Score (ranging from 21 to 35), most participants (85.9%) reported moderate adherence to the Mediterranean Diet. Further analysis examined the correlations of the PCS-12 score with the frequency of consumption of each of the 11 food categories, as well as all demographic and health behavior variables. The ranked correlations indicated a significant relationship between PCS-12 score and Med Diet adherence and the consumption of alcoholic beverages (r = 0.437, *p* < 0.05), in accordance with the Mediterranean Diet suggestions, as well as a negative relationship with BMI score (r = −0.454, *p* < 0.010). Moreover, significant correlations were also present between the physical quality of life and AQLQ score and work type, as well as gender, age, and marital status. The results of our study showed a high rate of obesity in patients with asthma at the General University Hospital of Larissa and moderate adherence to the Mediterranean Diet. Increased BMI and alcoholic beverage consumption in asthma patients were significant predictors of lower physical health-related quality of life. In conclusion, personal and society-level interventions are required to effectively address obesity and poor diet in patients with asthma.

## 1. Introduction

Western-style dietary patterns widely adopted in developed and developing countries are characterized by excessive energy intake, high consumption of processed or “fast foods”, and reduced consumption of fruits, vegetables, and whole grains [1]. The World Health Organization (WHO), as well as scientific and non-scientific organizations, place particular emphasis on the role of nutrition in the prevention of non-communicable diseases (NCDs) [2]. The role of nutrition in chronic diseases, such as cardiovascular diseases, type 2 diabetes, metabolic syndrome, and cancer, is considered indisputable, as well as an increase in the frequency of allergic diseases such as bronchial asthma with the adoption of a “Western lifestyle” [3,4].

According to the Global Initiative for Asthma (GINA), asthma is a common chronic airway disease that can be triggered at any age. Asthma symptoms are wheezing, shortness of breath, chest tightness, and coughing. The factors that may contribute to the development of bronchial asthma are low physical activity, long hours of sedentary life, and poor dietary patterns, as well as obesity, which may play an important role. According to the WHO, both children and adults with increased body weight and obesity are at greater risk of asthma [5,6].

Dietary recommendations aim to promote a healthy diet with lots of vegetables and fruits, to maintain a person’s ideal weight or to lose weight in obese patients [7].

According to the WHO, in 2019, asthma affected approximately 262 million people and caused 455,000 deaths and has been included in the WHO Global Action Plan for the Prevention and Control of NCDs and the United Nations 2030 Agenda for Sustainable Development [6]. Global estimates give ever-increasing rates of asthma, reaching 339 million people. If these increasing trends continue, the global prevalence of asthma is projected to reach 400 million people by 2025 [8]. From scientific data, it appears that bronchial asthma is more common in childhood. In adulthood, however, there appears to be a higher prevalence in women, as well as more severe asthma that is more common in women than in men [9].

The role of nutrition in the management of asthma in obese patients is of increasing interest due to their limited response to inhaled corticosteroids, as well as other difficulties faced by clinicians in the management of these patients [10]. Some studies note that through diet and lifestyle, there can be an improvement in asthma control [11]. Not only dietary patterns but also the period of exposure to these patterns, such as the prenatal period, childhood, and adolescence, as well as adulthood, play an important role in both the pathogenesis and progression of asthma. There is a correlation with the development of bronchial asthma in childhood if the mother during pregnancy has a reduced intake of vitamins E and D. Also, according to the study by Hancu and colleagues, asthma in adulthood is related to the diet followed by children during childhood; more specifically, a high consumption of highly processed foods and fast food in childhood leads to worse asthma control in adulthood [12]. Adopting the Mediterranean Diet (Med Diet), characterized by a high intake of whole grains, vegetables, fruits, olive oil, and fish, has been shown to have beneficial effects in controlling asthma symptoms [13,14].

Also, according to Alwarith J et al., lower socioeconomic status may be associated with a higher prevalence and higher mortality. This can be explained on the one hand as the lack of access of these patients to the medical community and medication and on the other hand by the purchase of foods with low nutritional value, such as foods high in fat and sugar, and the reduced consumption of fruits and vegetables [15].

According to the study by Wood et al., improvements in asthma patients were evident after an increased intake of fruits and vegetables [16]. Many food components have been proposed for their antioxidant, anti-allergic, and anti-inflammatory properties, which may protect against the risk of bronchial asthma [17]. Similar results are seen in the conclusions of the study by Alwarith J et al., which state that food components such as fatty acids (polyunsaturated, monounsaturated, saturated fatty acids), dietary fiber, and vitamin D can somehow affect the immune mechanisms of the body related to the pathophysiology of bronchial asthma [15].

Another factor considered necessary in patients with bronchial asthma is their functionality at each level of functioning, such as physical, individual, social, and mental health. According to the WHO, it is as important as the identification of the disease and whether the person can cope with their daily life, such as being able to work and carrying out usual activities at home, at work, at school, or in other social areas [18]. According to the Hellas Health VII: Pan-Hellenic survey on the health of the Greek population, carried out on a sample of 1001 people during 2017, 21% of respondents limited their daily activities or work obligations due to their physical health and 34% due to anxiety or melancholy [19]. The study by Leonie Burgess and colleagues on pregnant women with bronchial asthma showed that pregnant women might experience a worse quality of life when they cannot control their asthma symptoms [20].

This pilot study aimed to investigate the adherence to the Mediterranean Diet (Med Diet) and its association with asthma severity, quality of life, and functionality in patients with bronchial asthma.

## 2. Materials and Methods

A cross-sectional study was conducted at the General University Hospital of Larissa and, more specifically, on the patients of the outpatient asthma clinic. Those who gave their consent were included. Data collection started in March 2021 and ended in December 2022. The institutional ethics board approved the study. All procedures performed in this study followed the Declaration of Helsinki. The study protocol was registered and approved by the Ethics Committee of the Department of Medicine of the University of Thessaly (Larissa, 18.02.2021/Protocol number: 671).

Anthropometric measurements were taken, and the data collected were body weight and height using a Seca 7700-column mechanical scale/static meter. Body Mass Index (BMI) was calculated and classified according to the WHO into four categories: underweight (<18.5 kg/m^2^), average weight (18.5–24.9 kg/m^2^), overweight (25–29.9 kg/m^2^), and obesity (≥30 kg/m^2^) [21]. Information on physical activity, smoking, marital status, satisfaction with their financial situation, self-reported comorbidities, Asthma Control Test (ACT), Mediterranean Diet Score (Med Diet), Asthma Quality of Life Questionnaire (AQLQ), and Short Form Survey (SF12) questionnaires were also collected. The Mediterranean Diet Questionnaire is a tool that can be used to assess the adherence to the Mediterranean Diet, which is characterized by a high consumption of fruits, vegetables, whole grains, and olive oil, a moderate intake of fish and poultry, and limited consumption of red meat and processed foods. It also includes moderate amounts of red wine. The tool quantifies adherence to this dietary pattern, associated with numerous health benefits. Asthma severity classification was assessed by the hospital’s outpatient physician and based on medication use. The researchers officially requested the Med Diet and SF12 questionnaires, which are valid, reliable, and validated in the Greek language [22,23]. SF12 analysis was performed with the help of the tool “SF-12—OrthoToolKit” https://orthotoolkit.com/sf-12/ (accessed on 18 June 2023). Cronbach’s alpha for scales was for PCS-12 (Physical Score), a = 0.814, for MCS-12 (Mental Score), a = 0.741, for Med Diet Score, a = 0.643, and for AQLQ (32 items), a = 0.971.

### Statistical Analysis

Statistical analysis of the collected data was performed with ΙΒΜ SPSS v. 28. Demographic and health behavior data were reported with absolute and relative frequencies (*n*, %) for categorical variables and mean and standard deviation (M, SD) values for scale variables. The normality of scale variables was assessed with Shapiro–Wilk tests [24], indicating non-normal distributions. The outcome (dependent) variable of this study was the physical health-related quality of life (PCS-12). First, univariate analysis was performed using chi-square and Mann–Whitney tests between asthma severity (Mild = 1, Severe = 2) and demographics, health behavior, Mediterranean Diet adherence, and quality-of-life variables. Bivariate correlations between physical health-related quality of life and independent variables were performed with the non-parametric Spearman correlation coefficient. Multiple linear regression analysis was performed to examine the cumulative effect of Mediterranean Diet adherence, asthma-related quality of life, and BMI score on the physical health-related quality of life of asthma patients, controlling for demographic variables.

## 3. Results

The sample consisted of 85 participants (70.6% female), with a mean age of 57 years. More than half of the patients were married (63.9%). In total, 30.1% of participants reported working with manual tasks, 43.4% reported mixed work (manual and sedentary work), and 26.5% office work. Over 90% of participants reported low-to-moderate satisfaction with their financial situation. Regarding their physical activity, 43.5% reported low and 45% moderate physical activity. In total, 54.3% of the participants were non-smokers, while 93.4% had comorbidities. In relation to BMI, 12.9% of participants were of normal weight, 45.9% were overweight, 25.9% were obese level I, 5.9% were obese level II, and 9.4% were in the morbidly obese range. Based on the Med Diet Score (ranging from 21 to 35), most participants (85.9%) reported moderate adherence to the Mediterranean Diet. According to the descriptive statistics, the percentage of patients with severe asthma was 46% (receiving high dosages of ICSs according to the GINA definition), and the rate of patients with mild and moderate asthma (receiving mild and moderate dosages of ICSs, respectively) was 54%. Patients treated with biological therapy comprised 33% of the sample. In total, 78% were medicated with mepolizumab (nucala), 18% with omalizumab (xolair), and 4% with benralizumab (fasenra). Disease severity was determined by the dosages of medication, according to GINA 2022 guidelines [25].

As presented in Table 1, 39 participants (45.9%) were categorized as severe asthma patients. The severe asthma group had a significantly higher mean age (M = 52.37, SD = 18.10) than the mild asthma group (M = 62.46, SD = 11.89). Between the groups of severe and mild asthma patients, differences were detected in terms of the type of work (*p* = 0.040), where the severe asthma group reported higher levels of manual labor (43.6%) compared to mild asthma patients (18.2%). Interestingly, no substantial differences were discovered between the groups of mild and severe asthma patients in AQLQ score, health-related quality of life (PCS-12, MCS-12), or adherence to the Mediterranean Diet (i.e., Med Diet Score). Yet, the median score of physical health-related quality of life was lower in the severe asthma group (median = 43.66, range = 35.62) compared to the mild asthma group (median= 45.75, range = 37.31), for a significance level of 10% (*p* = 0.076).

Figure 1 presents the boxplots of PCS-12 and AQLQ z-scores for mild and severe asthma groups, indicating the pattern of lower PCS-12 for the severe asthma group compared to the higher PCS-12 score for the mild asthma group.

Intercorrelations between quality-of-life variables and the Mediterranean Diet Score are presented in Table 2. The physical health SF-12 score had a positive correlation with the mental health SF-12 score (r = 0.309, *p* < 0.010) and with the AQLQ score (r = 0.384, *p* < 0.010). Nevertheless, no significant correlation was detected between the physical SF-12 score and the aggregated Med Diet Score (*p* > 0.05). Further analysis examined the correlations of the PCS-12 score with the frequency of consumption of each of the 11 food categories, as well as all demographic and health behavior variables.

The ranked correlations presented in Figure 2 indicated a significant relationship between the PCS-12 score and Med Diet adherence and the recommended consumption of alcoholic beverages (r = 0.437, *p* < 0.05), more specifically the consumption of alcoholic beverages in accordance with the Mediterranean Diet suggestions, as well as a negative relationship with BMI score (r = −0.454, *p* < 0.010). Moreover, significant correlations were also present between the physical quality of life and AQLQ score and work type, as well as gender, age, and marital status. More likely to exhibit higher physical health-related quality of life were male participants who adhered to the Mediterranean Diet recommendation for alcohol consumption, had higher asthma-related quality of life, had mixed types of work, were single, younger, and had a lower BMI.

Multiple linear regression analysis was performed for physical quality of life (PCS-12), with the significantly correlated variables (adherence to the consumption of alcoholic beverages, BMI score, AQLQ score, work type, gender, age, marital status) as independent variables (R^2^ = 0.466, F(8, 71)= 7.74, *p* < 0.001). Physical health-related quality of life is positively predicted by asthma quality of life (β = 0.256, *p* = 0.008) and negatively predicted by BMI score (β = −0.415, *p* < 0.001), meaning that severe asthma-related quality of life and lower BMI scores lead to higher physical health-related quality of life. Moreover, adherence to alcoholic beverage consumption positively predicted physical health-related quality of life (r = 0.220, *p* = 0.027), i.e., asthma patients following Mediterranean alcohol consumption patterns were more likely to have better physical health (Table 3).

## 4. Discussion

This study aimed to investigate adherence to the Mediterranean Diet (Med Diet) and its association with asthma severity, quality of life, and functionality in patients with bronchial asthma. We found that a large percentage of 41.2% belonged to the category of obese patients. Almost half of the patients (45.9%) were categorized as severe asthma patients. According to Frontela-Saseta C et al., obesity in patients with asthma has been linked to increased systemic leukotriene inflammation, and the increase in adipose tissue may contribute to airway inflammation, thus exacerbating asthma symptoms [26].

In addition, overweight or obese individuals are more likely to develop asthma as a result of adipose tissue that is considered functionally active. Obesity causes inflammation and fibrosis of white adipose tissue, leading to local and systemic metabolic dysfunctions, and appears to contribute in some way to airway inflammation, lung function, and bronchial asthma exacerbations [27,28]. Obese adults tend to have more severe asthma than normal-weight adults and are also more likely to be hospitalized. In the United States, approximately 60% of adults with severe asthma are obese and have both poorer asthma control and poorer quality of life [29]. In addition, according to the results of the study by Maalej S and colleagues, obesity and overweight are associated with greater severity of asthma and worse control as well as lower quality of life of asthmatic patients. Obese patients with asthma had more frequent emergency room visits, more frequent hospitalizations, more days off work, and higher doses of medication. The relationship between obesity and bronchial asthma may be explained by chronic systemic inflammation [30].

Also, according to the review carried out by Rosser FJ et al., obesity is becoming more common in Hispanic America. In the United States, both overweight and obesity are more common in Mexican Americans and Puerto Ricans and are associated with asthma [31]. Overweight or obese adults are 50% more likely to develop asthma, with obesity accounting for 250,000 new cases of asthma each year in US adults [8].

Based on the Med Diet Score (ranging from 21 to 35), most participants (85.9%) reported moderate adherence to the Mediterranean Diet. Regarding adherence to the Med Diet, we observe moderate adherence in the majority of participants. It is possible that the results are due to the small number of participants or to the average age of those who were older. In other studies, also carried out in Greece but in a child population, the results showed that adherence to the Med Diet was negatively associated with asthma symptoms, or adherence to the Med Diet did not differ between the two control groups and there was no association with any of the three cytokines measured. Additionally, according to the review by Castro-Rodriguez et al., adherence to the Med Diet by children appears to have a protective effect on asthma symptoms and wheezing, but this effect is equivocal on lung function and bronchial hyperresponsiveness [32,33,34].

Interestingly, no substantial differences were discovered between the groups of mild and severe asthma patients in AQLQ score or health-related quality of life (PCS-12, MCS-12). Regarding the quality of life, which was tested with both questionnaires, the SF12 and the AQLQ, a mild impairment was observed in the quality of life of adult patients with severe asthma, as in the study by Apfelbacher CJ et al. [35]. This may be due to the fact that the patients were recruited through their fixed and scheduled appointments at the outpatient clinics of the Larissa University Hospital as well as the fact that they live in smaller cities. Living with family members and, in general, having friendly relationships which can be maintained better due to the short distance in kilometers can be linked to a better quality of life [36,37].

The only difference, with a statistical significance at a 0.10 level, was seen in the SF12 controlling for physical health, where the results showed lower physical health in the severe asthma group compared to the mild asthma group. Yet, the median score of physical health-related quality of life was lower in the severe asthma group compared to the mild asthma group. Significant correlations were also present between the physical quality of life and AQLQ score and work type, as well as gender, age, and marital status. Our results are consistent with the results of the study by Hyun Jin Song and colleagues, where patients with severe asthma had significantly lower PCS scores compared to those with mild asthma; in addition, factors such as gender, more specifically female gender, older age, and lower education were associated with poor physical health [38].

We found that there was a significant correlation between PCS-12 score and Med Diet adherence and the consumption of alcoholic beverages, meaning that asthma patients following Mediterranean alcohol consumption patterns were more likely to have better physical health. Moderate wine consumption is considered part of the Mediterranean Diet when consumed with food, and the recommendation is one glass per day for women and up to two glasses per day for men. Recommended wine consumption increases longevity and reduces the risk of cardiovascular disease [39]. Also, according to the study by Barbería-Latasa and colleagues, moderate consumption of red wine with meals and spread evenly throughout the week, avoiding excessive consumption, can reduce the risk of all-cause mortality by 48% [40]. Alcohol consumption is one of the main modifiable risk factors that contribute to the increase in non-communicable diseases, including chronic respiratory disease. Also, increased alcohol consumption significantly affects physical health and has been linked to increased BMI. The findings of the study by Ramos-Vera C and colleagues emphasize that frequent and occasional alcohol consumption can significantly affect physical health in a negative way [41]. Alcohol use disorder has been associated with increased healthcare utilization and the mortality of asthma patients. The management of alcohol consumption in asthmatics may improve disease and patient outcomes [42].

Physical health-related quality of life was negatively predicted by BMI score, meaning that lower BMI scores led to higher physical health-related quality of life. The results of the study by Gilles Louis and colleagues were similar, where BMI was associated with a lower score on the activity dimension of the AQLQ. In the results of the study by Goudarzi and colleagues carried out in Japan, it was more specifically shown that abdominal visceral fat was associated with reduced quality of life in asthmatic patients independently of other indicators of obesity [43]. Changes in BMI during adolescence have been associated with an increased risk of asthma, and monitoring these changes in this period of life may help in the primary prevention of asthma [44]. On the contrary, in a study carried out by Rhee H et al. in teenagers, while around 50% of the sample were obese or overweight, the negative effect of being overweight or obese on quality of life or asthma control has not yet been manifested in teenagers. The findings, according to the researchers of the study, highlight that adolescence is considered an ideal period for safe intervention to reduce excess body weight, which can prevent the potentially harmful effects of obesity on future asthma outcomes [45,46,47,48].

The limitations of this study are the small sample size and that the patients were recruited from a hospital in Greece. Also, due to the type of study, “cross-sectional study,” we cannot provide evidence of cause and effect. Finally, there was a greater participation by women than men, as well as an age difference between the severe and mild asthma groups.

## 5. Conclusions

The results of our study show a high rate of obesity in patients with asthma at the General University Hospital of Larissa. Obesity is considered a major risk factor for asthma and is also associated with increased asthma severity. Increased BMI and alcoholic beverage consumption in asthma patients were significant predictors of lower physical health-related quality of life. Moderate adherence to the Mediterranean Diet and a relatively good level of quality of life of the patients were also observed. Intervention is required at both the individual and societal levels to address both obesity and asthma in obese individuals, so we can say that effective public health interventions are necessary.

## Figures and Tables

**Figure 1 jpm-13-01512-f001:**
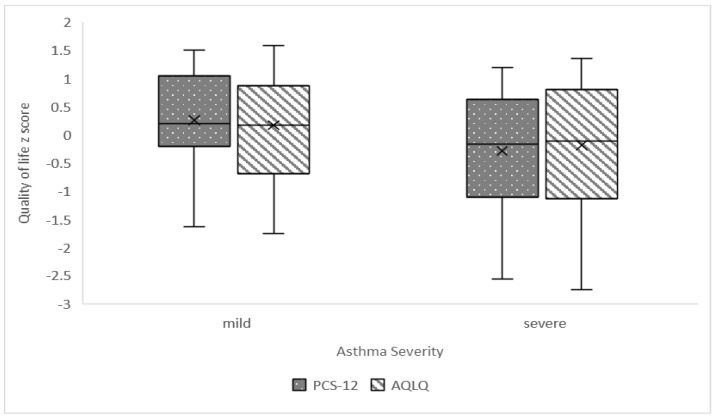
Boxplots of PCS-12 and AQLQ z-scores for mild and severe asthma groups.

**Figure 2 jpm-13-01512-f002:**
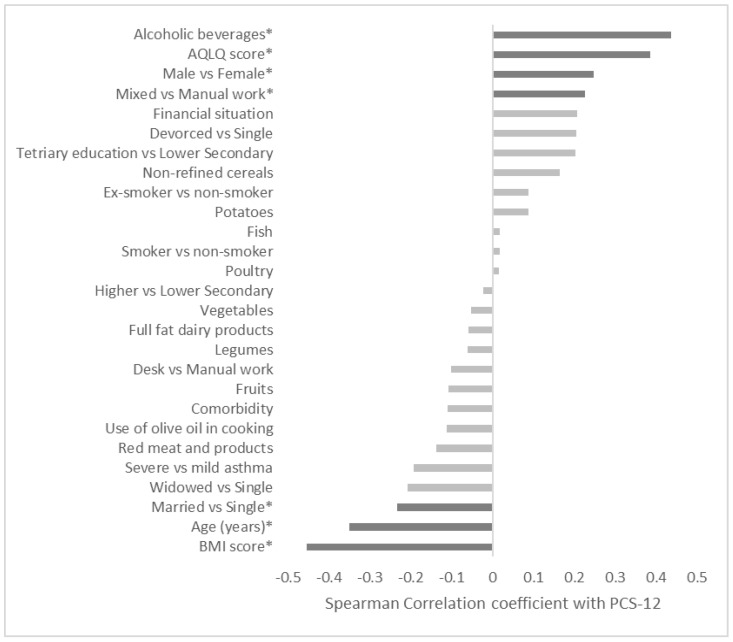
Ranked correlations of PCS-12 score with each item of the 11 food categories and all demographic and health behavior variables. Variables indicated with (*) presented statistically significant correlations with the PCS-12 score.

**Table 1 jpm-13-01512-t001:** Demographic characteristics of the sample and differences based on asthma severity.

		Total (*n* = 85)		Asthma Severity (GINA)				
				Mild (*n* = 46)		Severe (*n* = 39)		
		*n*	%	*n*	%	*n*	%	*p* ^1^
Gender	Men	25	29.4	10	21.7	15	38.5	0.092
	Women	60	70.6	36	78.3	24	61.5	
Educational level	Lower secondary	32	37.6	15	32.6	17	43.6	0.161
	Higher secondary	22	25.9	10	21.7	12	30.8	
	Tertiary education	31	36.5	21	45.7	10	25.6	
Work type	Manual	25	30.1	8	18.2	17	43.6	0.040
	Mixed	36	43.4	23	52.3	13	33.3	
	Desk	22	26.5	13	29.5	9	23.1	
Physical activity	Low	37	43.5	16	34.8	21	53.8	0.206
	Moderate	39	45.9	24	52.2	15	38.5	
	High	9	10.6	6	13.0	3	7.7	
Economic situation	Low satisfaction	39	46.4	21	45.7	18	47.4	0.062
	Moderate	38	45.2	24	52.2	14	36.8	
	High	7	8.3	1	2.2	6	15.8	
Marital status	Single	15	18.1	13	28.9	2	5.3	0.034
	Married	53	63.9	24	53.3	29	76.3	
	Divorced	9	10.8	5	11.1	4	10.5	
	Widower	6	7.2	3	6.7	3	7.9	
Smoking	Smoker	8	9.9	5	11.1	3	8.3	0.343
	Non-smoker	44	54.3	27	60.0	17	47.2	
	Former smoker	29	35.8	13	28.9	16	44.4	
BMI category	Normal weight	11	12.9	10	21.7	1	2.6	0.080
	Overweight	39	45.9	19	41.3	20	51.3	
	Obesity Level I	22	25.9	10	21.7	12	30.8	
	Obesity Level 2	5	5.9	2	4.3	3	7.7	
	Pathogenic obesity	8	9.4	5	10.9	3	7.7	
Comorbidities	Yes	57	93.4	29	93.5	28	93.3	0.973
	No	4	6.6	2	6.5	2	6.7	
Age (years) (mean (SD))		57 (16.28)		52.37 (18.10)		62.46 (11.89)		0.011
BMI score ^2^ (mean (SD))		30 (6.29)		29.74 (6.85)		31.16 (5.47)		0.122
PCS-12 ^3^ (median (range))		45.09 (37.31)		45.75 (37.31)		43.66 (35.62)		0.076
MCS-12 ^4^ (median (range))		47.23 (40.42)		46.51 (38.14)		48.37 (39.02)		0.954
Med Diet Score ^5^ (median (range))		28 (20)		28 (20)		28 (19)		0.586
AQLQ score ^6^ (median (range))		166 (224)		170 (224)		163 (216)		0.500

^1^ *p* < 0.01, ^2^ BMI: Body Mass Index, ^3^ PCS-12: Physical Score, ^4^ MCS-12: Mental Score, ^5^ Med Diet Score: Mediterranean Diet Score, ^6^ AQLQ: Asthma Quality of Life Questionnaire.

**Table 2 jpm-13-01512-t002:** Descriptive statistics and Spearman correlation coefficients between quality-of-life variables and Mediterranean Diet Score.

	M ^2^	SD ^3^	Median	IQR ^4^	Shapiro–Wilk	1	2	3	4
PCS-12	44.02	9.41	45.09	12.23	0.001	--			
2. MCS-12	45.26	10.74	47.23	17.44	0.004	0.309 ^1^	--		
3. Med Diet Score	28.85	4.34	28.00	5.00	0.044	−0.114	0.026	--	
4. AQLQ score	157.82	51.19	166.00	66.00	<0.001	0.384 ^1^	0.183	0.013	--

^1^ p: *p* < 0.01, ^2^ Μ: mean value, ^3^ SD: standard deviation, ^4^ IQR: interquartile range.

**Table 3 jpm-13-01512-t003:** Regression coefficients for the predictors of physical quality of life.

	B	SE	β	*p*
Med Diet adherence to alcoholic beverages	1.102	0.487	0.220	0.027
AQLQ score	0.047	0.017	0.256	0.008
BMI score	−0.627	0.144	−0.415	<0.001
Gender (male vs. female)	2.225	1.940	0.108	0.255
Age (years)	−0.044	0.080	−0.074	0.586
Marital status				
Married vs. single	−1.088	3.276	−0.055	0.741
Divorced vs. single	2.543	3.654	0.085	0.489
Widowed vs. single	−3.480	4.710	−0.097	0.463
Work type				
Mixed vs. manual work	2.208	2.064	0.116	0.288
Desk vs. manual work	1.330	2.411	0.063	0.583

## Data Availability

The data presented in this study are available on request from the corresponding author. The data are not publicly available due to privacy reasons.

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
