# Peer review of "Physical Health-Related Quality of Life in Relation to Mediterranean Diet Adherence in a Sample of Greek Asthma Patients: A Pilot Study"

_jpm, 2023, doi:10.3390/jpm13101512_

Round 1

Reviewer 1 Report

This is an interesting pilot study addressing the mutual dependencies between diet, asthma and quality of life issues.

The study is cross-sectional which may be considered one of its drawbacks, however, the provided data are promising and will hopefully be expanded in the future.

Please find below my comments regarding the manuscript.

Regarding the study group: there is a considerably high percentage of severe asthma patients. Is this intentional, or were the consecutive patients recruited? The authors do not mention whether they want to continue the study on larger group which would be more evenly distributed with regard to asthma severity. Please comment on that.

Data about the treatment are not provided. Have the subjects been treated with accordance to GINA or Greek national guidelines? If possible, providing data on mean inhaled GCS dose and other asthma treatments would render the picture more detailed and might add new context to the QoL data. Were the severe asthma patients treated with biologicals also included? I understand this is a preliminary study, but this should be at least addressed in the discussion and commented, or more complete data provided.

Do the Authors have data on some metabolic parameters which would reflect the adherence to Med diet and give an insight to the intensity of the inflammatory process, e.g., serum lipid levels, blood glucose, HbA1c, CRP and/or other?

Regarding methods: I would suggest providing some details about Mediterranean diet adherence questionnaire – not all allergists may be familiar with the tool and its interpretation.

Some minor issues

Lines 29-30 in abstract and  line 166 in the manuscript text: this sentence is a little bit confusing, please clarify which correlations were exactly observed; please explain what does the phrase „a significant relationship between PCS-12 score and med. Diet adherence to the consumption of alcoholic beverages” mean. Does it apply to Mediterranean pattern pf alcoholic beverages consumption? I suppose it is so, but please clarify becuase this may look not fully understandable.

Lines 216-218: Why do the Authors associate decrease of the quality of life with assessment during scheduled appointments and living in smaller cities? Please provide more detailed explanation.

Overall, this is an interesting pilot study.

No significant remarks regarding English language quality. Minor editing, preferentially by a native speaker, recommended.

Reviewer 2 Report

First of all, I appreciate the opportunity to review this interesting article that deals with the relationship between adherence to the Mediterranean diet in asthmatic patients and improvement in quality of life.

In general, it seems to me that it is an interesting article that relates aspects described in low frequency in the literature, with eating habits being one of the easily modifiable variables, which are strongly related to the state of health in general, especially, frequency and style of alcohol consumption.

The article describes in detail its methodology, makes a good introduction to the problem and its conclusions and discussion are in accordance with the results obtained. In general, it seems to me that in its current state it is of great interest, even with the limitations mentioned by the authors in terms of sample size and study design, I believe that it can serve as a preamble to encourage more studies of this type on a large scale.

As my only suggestion, I advise, in the abstract, to add a little of what corresponds to the authors' conclusions and not just the results.
